# Characterization of Recurrent Relevant Genes Reveals a Novel Role of RPL36A in Radioresistant Oral Squamous Cell Carcinoma

**DOI:** 10.3390/cancers13225623

**Published:** 2021-11-10

**Authors:** Ting-Wen Chen, Kai-Ping Chang, Chun-Chia Cheng, Cheng-Yi Chen, Shu-Wen Hong, Zong-Lin Sie, Hsing-Wen Cheng, Wei-Chen Yen, Yenlin Huang, Shu-Chen Liu, Chun-I Wang

**Affiliations:** 1Institute of Bioinformatics and Systems Biology, National Yang Ming Chiao Tung University, Hsinchu 300, Taiwan; Dodochen@nctu.edu.tw; 2Department of Biological Science and Technology, National Yang Ming Chiao Tung University, Hsinchu 300, Taiwan; 3Center for Intelligent Drug Systems and Smart Bio-Devices (IDS2B), National Yang Ming Chiao Tung University, Hsinchu 300, Taiwan; 4Department of Otolaryngology-Head & Neck Surgery, Chang Gung Memorial Hospital, Taoyuan 333, Taiwan; dr.kpchang@gmail.com (K.-P.C.); aimee10221@gmail.com (H.-W.C.); d56610@yahoo.com.tw (W.-C.Y.); 5College of Medicine, Chang Gung University, Taoyuan 333, Taiwan; 6Molecular Medicine Research Center, Chang Gung University, Taoyuan 333, Taiwan; 7Radiation Biology Research Center, Institute for Radiological Research, Chang Gung University, Taoyuan 333, Taiwan; cccheng.biocompare@gmail.com (C.-C.C.); g8935677@gmail.com (S.-W.H.); zonlins@gmail.com (Z.-L.S.); 8Department of Cell Biology and Anatomy, College of Medicine, National Cheng Kung University, Tainan 701, Taiwan; cychen@gs.ncku.edu.tw; 9Department of Pathology, Chang Gung Memorial Hospital, Taoyuan 333, Taiwan; louisyhuang@gmail.com; 10Department of Biomedical Sciences and Engineering, National Central University, Taoyuan 333, Taiwan

**Keywords:** oral cancer, oral cavity squamous cell carcinoma, OSCC, radioresistance, RPL36A, radiotherapy

## Abstract

**Simple Summary:**

Radioresistance is one of the major factors contributing to radiotherapy failure in OSCC. By systematically comparing the prognostic values of all genes in TCGA-OSCC patients with and without radiotherapy, radioresistance-associated genes were identified. Higher RPL36A transcript levels were found to be associated with a poor prognosis only in OSCC patients with radiotherapy in the cohort of TCGA and another independent Taiwanese cohort. RPL36A was then shown to be involved in the regulation of DNA damage, cell cycle and apoptosis, leading to radioresistance. Thus, such integrated studies are expected to be greatly beneficial for the development of new therapeutic interventions for radioresistant OSCC in the future.

**Abstract:**

Radioresistance is one of the major factors that contributes to radiotherapy failure in oral cavity squamous cell carcinoma (OSCC). By comparing the prognostic values of 20,502 genes expressed in patients in The Cancer Genome Atlas (TCGA)-OSCC cohort with (*n* = 162) and without radiotherapy (*n* = 118), herein identified 297 genes positively correlated with poor disease-free survival in OSCC patients with radiotherapy as the potential radioresistance-associated genes. Among the potential radioresistance-associated genes, 36 genes were upregulated in cancerous tissues relative to normal tissues. The bioinformatics analysis revealed that 60S ribosomal protein L36a (RPL36A) was the most frequently detected gene involved in radioresistance-associated gene-mediated biological pathways. Then, two independent cohorts (*n* = 162 and *n* = 136) were assessed to confirm that higher RPL36A transcript levels were significantly associated with a poor prognosis only in OSCC patients with radiotherapy. Mechanistically, we found that knockdown of RPL36A increased radiosensitivity via sensitizing cells to DNA damage and promoted G2/M cell cycle arrest followed by augmenting the irradiation-induced apoptosis pathway in OSCC cells. Taken together, our study supports the use of large-scale genomic data for identifying specific radioresistance-associated genes and suggests a regulatory role for RPL36A in the development of radioresistance in OSCC.

## 1. Introduction

Oral cavity squamous cell carcinoma (OSCC) is the most common cancer among head and neck squamous cell carcinomas (HNSCCs); it accounts for approximately 3% of all newly diagnosed cancer cases [1]. Generally, the standard treatment for OSCC patients is primary excision alone for patients at an early stage or excision by surgery followed by adjuvant radiotherapy or concurrent chemoradiotherapy (CCRT) for late and advanced metastatic cancers [2]. Tumor recurrence after radiotherapy is a major obstacle to complete curability in OSCC [2], and a high recurrence rate is still one of the major problems in OSCC treatment [3]. Although various drugs have been proposed for the systemical or topical treatment of OSCC, no current therapies seem to have significant effects on a patient’s prognosis [4,5] The overall relative 5-year survival rate of OSCC is approximately 60%, but the survival rates vary among the different stages. Camisasca et al. reported that the 5-year survival rate was approximately 90% in OSCC patients without recurrence and 30% in patients with recurrence [6]. Early-stage OSCC disease is treated with relatively good outcomes; however, nearly 65% of patients present with advanced disease (stage III and IV), and fewer than 30% of these patients are cured. High failure rates and low survival rates are observed in patients undergoing CCRT with recurrent, intractable OSCC. More than 30% of patients eventually develop local recurrence or metastasis, usually within the first 2 years of follow-up after OSCC treatment [7].

Radioresistance is one of the major factors related to radiotherapy failure in OSCC. The predisposing factors and a corresponding prediction model have not yet been fully elucidated. Several studies have established radioresistant OSCC cell lines to identify genes with altered expression in response to radioresistance [8]. Most researchers are now focusing on characterizing the underlying molecular mechanisms in an attempt to target specific genes or pathways involved in radioresistant OSCC tumors. Recently, different experimental approaches have been adopted, including the direct comparison of two sets of samples with different levels of radiosensitivity from cancer tissues or cell lines [9]. Chen et al. found that the knockdown of LINC00662 repressed AK4 and attenuated radioresistance in OSCC cell lines [10]. Lin et al. showed that GP96 was overexpressed in OSCC cell lines and was a poor prognostic indicator for patients receiving radiotherapy [11]. Li et al. demonstrated that GDF15 contributed to radioresistance and cancer stemness by regulating cellular reactive oxygen species (ROS) levels via a SMAD-associated signaling pathway in OSCC cell lines and a mouse model [12]. Notably, epidermal growth factor receptor (EGFR) is a receptor tyrosine kinase that is highly expressed in OSCC and is a common target in various cancer types [13]. EGFR overexpression is observed in more than 80% of HNSCCs. An anti-EGFR monoclonal antibody against the extracellular ligand-binding domain of the receptor has shown promising results as an adjuvant therapy in several solid tumor types, including OSCC [14]. Cetuximab, an EGFR-blocking antibody, has also been shown to enhance the cytotoxic effects of radiation in squamous cell carcinoma of the head and neck [15]. Accordingly, the identification of the radioresistance-associated molecules that contribute to a poor prognosis may facilitate patient consultation to determine proper treatment selection to improve the therapeutic outcome. However, there are limited studies that have used publicly available databases that contain large sample sizes and have systematically analyzed radioresistance-associated genes.

Therefore, we compared the prognostic values of 20,502 genes in patients in The Cancer Genome Atlas (TCGA)-OSCC cohort with (*n* = 162) and without radiotherapy (*n* = 118) and identified 297 genes positively correlated with poor disease-free survival (DFS) in OSCC patients with radiotherapy as the potential radioresistance-associated genes. Using this strategy, specific radioresistance-associated gene candidates were identified, and we validated that 60S ribosomal protein L36a (RPL36A) is a promising radioresistance-associated biomarker in the Taiwanese population. In the current study, we aimed to elucidate the clinical and biological significance of RPL36A in radioresistant OSCC.

## 2. Materials and Methods

### 2.1. Patient Populations and Clinical Specimens

In this study, for the testing cohort, 136 patients were enrolled, whose untreated OSCC tumors had been primarily managed by surgical resection with subsequent radiotherapy or CCRT. Tumor specimens and pericancerous normal tissues for real-time quantitative PCR (qPCR) analysis were surgically resected from patients diagnosed with OSCC who were consecutively selected from 2006 to 2013. The patients included in this study underwent standard preoperative assessments according to institutional guidelines, including a detailed medical history assessment, complete physical examination, computed tomography or magnetic resonance imaging scans of the head and neck, chest radiographs, bone scans, and abdominal ultrasound. Primary tumors were excised with adequate margins under intraoperative frozen section control. After surgery, the pathological TNM classification of all tumors was performed according to the American Joint Committee on Cancer Staging Manual (2010). After discharge, all patients participated in regular follow-up visits every 2 months in the first year, every 3 months in the second year, and every 6 months thereafter.

### 2.2. Identification of Specific Radioresistance-Associated Genes for OSCC from the TCGA Dataset

Among the 315 OSCC patients, only 280 patients had available radiotherapy treatment information and available clinical survival data; this group included 162 patients with radiotherapy and 118 patients without radiotherapy. Messenger RNA expression levels and survival information for the TCGA-HNSCC dataset were downloaded from Broad GDAC Firehose (https://gdac.broadinstitute.org, TCGA data version 28 January 2016). Then, we examined the association between the DFS of the two cohorts and expression levels. For each gene, the mRNA expression values calculated with RNA-Seq data by expectation maximization (RSEM) [16] were used to stratify the patients in the two cohorts into two equal-sized groups per cohort. Kaplan–Meier survival plots and log-rank tests were used for the comparison of survival data between the high expression and low expression groups.

### 2.3. Core Molecular Pathways and Gene Ontology (GO) Analysis

The enriched GO terms and Kyoto Encyclopedia of Genes and Genomes (KEGG) pathways of the radioresistance- and RPL36A-associated genes were analyzed using the Database for Annotation, Visualization, and Integrated Discovery (DAVID, http://david.abcc.ncifcrf.gov/, accessed on 6 August 2020). The pathways in the top 10 with a *p*-value < 0.05 were selected as statistically significant pathways.

### 2.4. Identification of Differentially Expressed Genes (DEGs) in OSCC from the TCGA Dataset

We analyzed 315 OSCC tumor samples and 30 normal samples from the TCGA. Transcripts per kilobase million (TPM) values representing mRNA expression calculated by RSEM were used for DEG detection with Partek Genomics Suite software (Inc. P. Partek Genomics Suite, St. Louis, MO, USA).

### 2.5. Analysis of the Relationship between RPL36A and Other Genes in the TCGA-OSCC Dataset

Spearman’s correlations between the expression levels of RPL36A and all the other genes were calculated, followed by adjustment of the *p*-value with the Benjamini-Hochberg method. Spearman’s correlation coefficient greater than 0.2 was used as the cutoff value.

### 2.6. Cell Culture

The SAS cells (Prof Chien, Chang Gung University) were maintained in Dulbecco’s modified Eagle’s medium (DMEM) (Invitrogen, Carlsbad, CA, USA) containing 10% fetal bovine serum (FBS) (Gibco BRL, Carlsbad, MD, USA) plus antibiotics. The KOSC3 cells (Dr Chu, Chang Gung University) and the OEC-M1 cells (Prof Tsai, Chang Gung University) were cultured in Roswell Park Memorial Institute medium 1640 (Invitrogen) containing 10% FBS plus antibiotics. The cells were cultured at 37 °C in a humidified atmosphere of 95% air and 5% CO_2_.

### 2.7. Gene Knockdown of RPL36A Using Small Interfering RNA

Briefly, 19-nucleotide RNA duplexes targeting human RPL36A were synthesized and annealed by Dharmacon (Thermo Fisher Scientific, Rockford, IL, USA). The SAS and KOSC3 cells were transfected with the Dharmacon ON-TARGETplus Nontargeting Control Pool (Thermo Fisher Scientific), which contained a pool of four siRNAs designed and tested for minimal targeting of human, mouse, or rat genes (UGGUUUACAUGUCGACUAA, UGGUUUACAUGUUGUGUGA, UGGUUUACAUGUUUUCUGA, UGGUUUACAUGUUUUCCUA) or RPL36A-pooled siRNA (GAGGAAGUGCGAUCGGGAA, GCCGAUAGCGCUCACGCAA, GGGCAAACUAAGCCGAUUU, CCAUAAAGUGACACAGUAC) using Lipofectamine RNAiMAX reagents (Invitrogen), according to the manufacturer’s protocol. After 48 h of transfection, cell lysates were prepared for Western blotting to determine gene knockdown efficacy. The uncropped western blot figures can be found in Appendix A.

### 2.8. RNA Extraction and qPCR

The OSCC tumor and normal counterpart tissues were homogenized in liquid nitrogen with a mortar and pestle and incubated with RNAzol B reagent (Tel-Test, Friendwood, TX, USA). The RNA was further purified using a RNeasy cleanup kit (Qiagen, Germantown, MD, USA), according to the manufacturer’s protocol. First-strand cDNA was synthesized from 5 μg of total RNA, and then mixed with a reaction mixture consisting of commercially available primers (RPL36A, Hs01586542_g1 and normalization control ACTB, Hs01060665_g1, Assay-on-Demand, Applied Biosystems, Foster City, CA, USA), RNase-free water, and TaqMan Universal PCR Master Mix. The qPCR was performed and analyzed using a 7900 HT Sequence Detection System and SDS version 2 (Applied Biosystems). All experiments were repeated in duplicate.

### 2.9. Flow Cytometry Analysis

The cells were transfected with control siRNA or RPL36A siRNA for 24 h, followed by irradiation treatment for 48 h, were harvested by trypsinization. For cell cycle analysis, cells were fixed in 70% ice-cold ethanol comprising 2 mg/mL RNase for 30 min and were finally stained with propidium iodide (PI, 50 mg/mL) for 10 min. The fluorescence of PI in transfected cells was determined via flow cytometry analysis (FACScan System, Becton Dickinson, San Diego, CA, USA). We counted the percentage of cells in the sub-G1, G0/G1, S, and G2/M phases using CellQuest programs. For apoptosis detection, a commercial kit containing annexin V-FITC and PI was used (Strong Biotech Corporation, Taipei, Taiwan). The cells were analyzed using a FACSCalibur flow cytometer (BD Bioscience, San Jose, CA, USA) after annexin V-FITC and PI staining.

### 2.10. Clonogenic Survival Assay

The transfected cells were plated into 6-well plates (300 cells/well). After culture for 24 h, the cells were subjected to irradiation (0, 4, or 6 Gy). For colony formation, the irradiated cells were cultured for an additional 7 days. The colonies were fixed with acetic acid/methanol solution at a 1:3 ratio, and then stained with 0.5% crystal violet solution. The number of colonies (>50 cells/colony) was counted and analyzed using a Lionheart FX automated microscope (BioTek, Winooski, VT, USA). The surviving fraction was calculated by dividing the plating efficiency of the irradiation groups by the plating efficiency of the appropriate nonirradiated control.

### 2.11. Statistical Analysis

The Wilcoxon test was used to analyze the qPCR results from OSCC and normal counterpart tissues. The Cox proportional hazards regression was used to examine the hazard ratio for potential risk factors, including age, sex, smoking status, pathological stage, and N classification. Among the 162 OSCC patients with radiotherapy in the TCGA cohort, 147 patients had clinical information for all risk factors of interest which was used for the Cox regression analysis. The Cox regression, hazard ratio estimation, and 95% confidence intervals for hazard ratios were calculated using the R package survival analysis. A comparison of survival rates was carried out with the log-rank test, and the results were visualized with Kaplan–Meier plots. All patients underwent follow-up evaluations at our outpatient clinic until December 2016 or their death. Statistical analyses were performed using GraphPad Prism V5.01 (GraphPad Software, Inc., San Diego, CA, USA). Student’s *t*-test was used to compare two groups. Two-tailed *p*-values of 0.05 or less were considered statistically significant.

## 3. Results

### 3.1. Identification of Specific Radioresistance-Associated Genes in OSCC from the TCGA Dataset

In order to identify the specific radioresistance-associated genes, we analyzed the prognostic values of 20,502 genes in 162 TCGA-OSCC patients with radiotherapy and 118 TCGA-OSCC patients without radiotherapy. We found that 355 and 1441 genes were positively associated with poor DFS (*p* < 0.05) in 162 and 118 OSCC patients, respectively. Through comparison of the two gene groups, herein, we identified 297 genes specifically in OSCC patients with radiotherapy and positively correlated with poor DFS as the potential radioresistance-associated genes (hereafter referred to as OSCC-TCGA-RT-Risk) (Figure 1A).

### 3.2. Core Molecular Pathway and Gene Ontology Analyses in OSCC-TCGA-RT-Risk

To obtain a global picture of the molecular pathways that may contribute to radioresistance, the 297 radioresistance-associated genes in the OSCC-TCGA-RT-Risk dataset were analyzed with DAVID bioinformatics resources [17] to identify enriched KEGG or GO pathways. Significant GO term annotation by gene set enrichment analysis (GSEA) showed that the 297 radioresistance-associated genes primarily participated in the following categories: SRP-dependent co-translational protein targeting to membrane, translation, nuclear-transcribed mRNA catabolic process, nonsense-mediated decay, viral transcription, rRNA processing, translational initiation, mitochondrial translational elongation, mitochondrial translational termination, mRNA splicing via spliceosome, mitochondrial electron transport, and NADH to ubiquinone (Figure 1B). The KEGG pathway analysis showed enrichment in the following categories: ribosome, oxidative phosphorylation, Parkinson’s disease, Huntington’s disease, Alzheimer’s disease, nonalcoholic fatty liver disease (NAFLD), spliceosome, nucleotide excision repair, metabolic pathway, and systemic lupus erythematosus (Figure 1C). These results suggest a widespread impact of radioresistance-associated genes on translation-related pathways.

### 3.3. Selection of Upregulated Radioresistance-Associated Genes in OSCC Tumors

To identify novel radioresistance-associated genes that were upregulated in tumors and correlated with a poorer survival of patients with OSCC, first, we analyzed 315 OSCC tumors and 30 normal samples from TCGA. A total of 2375 DEGs that were upregulated in tumors relative to normal tissues (hereafter referred to as OSCC-TCGA-DEG-Up) were detected according to the criteria of an adjusted *p-*value < 0.05 and a fold change > 2. Accordingly, we compared the OSCC-TCGA-RT-Risk and OSCC-TCGA-DEG-Up groups (Figure 1A). Through this comparison, we found 36 potential gene candidates that were upregulated in tumors as compared with normal tissue and potentially associated with radioresistant OSCC. Among these candidates, 14 candidates have been previously reported to be dysregulated in HNSCC, and 7 candidates have been reported to be involved in radioresistance via a literature search (Table 1). Collectively, these results support the practical feasibility of the strategy, proposed in this study, for identifying radioresistance-associated gene candidates in cancers.

### 3.4. High RPL36A Expression Predicts a Poor Prognosis and Radioresistance in OSCC Patients

As shown in Table 1, we identified 22 novel radioresistance-associated genes that have not been reported to be dysregulated in OSCC. To choose promising candidates for further study, we focused on reviewing the identified radioresistance-associated genes in the significant GO and KEGG pathways of the OSCC-TCGA-RT-Risk. The bioinformatics analysis revealed that RPL36A was the most frequently detected gene involved in radioresistance-associated genes-mediated biological pathways (Appendix A). Importantly, the prognostic significance and radioresistance-related function of RPL36A in OSCC cells has not been elucidated. As shown in Figure 2A, high RPL36A transcript levels presented prognostic value only in OSCC patients with radiotherapy and not in without radiotherapy according to the OSCC-TCGA dataset. To determine whether high RPL36A expression is an independent prognostic factor for DFS, a multivariate analysis was additionally performed. The results showed that patients with advanced node classification (hazard ratio = 2.055 and *p* = 0.0054) and high RPL36A expression (hazard ratio = 1.980 and *p* = 0.0060) had significantly lower DFS after adjusting for age, sex, smoking status, and overall TNM stage (Table 2). These results confirmed that high RPL36A expression is an independent prognostic factor of DFS. Next, we assessed whether RPL36A expression levels were associated with OSCC patient responses to radiotherapy using a validation-testing Taiwanese cohort, including 136 OSCC tissue specimens from tumors and corresponding normal tissue specimens. All patients were primarily managed by surgical resection and subsequently underwent radiotherapy or CCRT. We found that the mRNA levels of RPL36A were increased in OSCC tumors as compared with the corresponding normal tissues (Figure 2B). the Kaplan–Meier curves and log-rank tests suggested that patients with high RPL36A expression exhibited poorer DFS rates than those with low RPL36A expression, and the 5-year DFS rates of patients with high and low RPL36A expression were 51.5% and 67.7%, respectively (*p* = 0.0477, Figure 2C). The results of multivariate analysis also confirmed that patients with advanced node classification and high RPL36A expression had significantly lower DFS in validation testing with the Taiwanese cohort (hazard ratios 2.262 and 1.782; *p* = 0.008 and 0.042, respectively, Appendix A).

### 3.5. Core Molecular Pathways and Gene Ontology Are Similar between Radioresistance- and RPL36A-Associated Genes

To obtain a comprehensive profile of molecules associated with RPL36A, we calculated the correlations between the expression of RPL36A and all the other genes in the OSCC-TCGA dataset. Using a Spearman’s correlation coefficient greater than 0.2 as the cutoff value, we identified 1043 and 2690 genes that were positively and negatively correlated with RPL36A, respectively (Figure 3A). We further analyzed the 1043 positively correlated genes using the DAVID bioinformatics resource to identify enriched KEGG or GO pathways. Significant GO term annotation by GSEA showed that the 1043 genes participated primarily in the following categories: translation, rRNA processing, SRP-dependent co-translational protein targeting to membrane, translational initiation, viral transcription, nuclear-transcribed mRNA catabolic process, nonsense-mediated decay, mitochondrial translational elongation, mitochondrial translational termination, mitochondrial respiratory chain complex I assembly, and anaphase-promoting complex-dependent catabolic process (Figure 3B). The KEGG pathway analysis showed enrichment in the following categories: ribosome, oxidative phosphorylation, Huntington’s disease, Parkinson’s disease, Alzheimer’s disease, NAFLD, pyrimidine metabolism, proteasome, ribosome biogenesis in eukaryotes, and spliceosome (Figure 3C). Taken together, these results suggest that the core molecular pathways and gene ontology are similar between the genes positively associated with RPL36A and radioresistance-associated genes.

### 3.6. RPL36A Depletion Sensitizes Cell to DNA Damage and Promotes G2/M Cell Cycle Arrest in Response to Irradiation

According to the GO and KEGG pathway analyses, we observed that RPL36A-associated molecules were highly involved in translation-related pathways. A previous study indicated that translational regulation and protein synthesis play fundamental roles in cell cycle progression [40]. On the basis of this concept, we proposed that RPL36A knockdown may lead to dysregulation of the cell cycle distribution and that a synergistic effect would be observed in response to irradiation. Therefore, we adopted a siRNA approach to suppress the expression of endogenous RPL36A in SAS, OEC-M1, and KOSC3 cells and assessed the effects on the cell cycle. A representative result is shown in Figure 4A. As compared with control siRNA-transfected cells, the percentages of cells in the G2/M phase significantly increased by 7.6% (*p* = 0.0406), 32.0% (*p* < 0.0001), and 21.9% (*p* = 0.0018) in RPL36A-knockdown SAS, OEC-M1, and KOSC3 cells, respectively, after 48 h of 6 Gy irradiation treatment (Figure 4B). To confirm these findings, we further determined changes in phosphorylation of H2AX (γ-H2AX), as a sensor of double-strand breaks and total and phosphorylated Chk1 and Chk2, which are involved in the G2/M checkpoint pathway. As shown in Figure 4C, the levels of γ-H2AX and Chk2 phosphorylation (Thr68) were remarkably increased in RPL36A-knockdown SAS, OEC-M1, and KOSC3 cells as compared with those in the control cells after irradiation treatment. These results indicated that knockdown of RPL36A in OSCC cells affected the radiosensitive phenotype via sensitizing cell to DNA damage and activated Chk2, thereby, leading to G2/M cell cycle arrest.

### 3.7. RPL36A Depletion Contributes to Radiosensitivity by Augmenting Irradiation-Induced Apoptosis

Because the knockdown of RPL36A sensitizes cell to DNA damage, followed by G2/M arrest in OSCC cells, the proportion of sub-G1 cells, which is an indicator of apoptosis, also increased in RPL36A-knockdown cells upon irradiation (Figure 4). Notably, previous studies have indicated that irradiation can induce apoptosis in many types of cancer, such as osteosarcoma, lung cancer, OSCC, and HNSCC [41,42,43,44,45,46]. To further assess the apoptosis associated with RPL36A knockdown-induced radiosensitization, annexin V/PI flow cytometry assays were applied. Figure 5A (up) shows the representative results. The results of apoptosis quantification shown in Figure 5A (down) revealed that the proportions of apoptotic cells significantly increased by 7.2%, 1.7%, and 6.0% in RPL36A-knockdown SAS, OEC-M1, and KOSC3 cells, respectively. Importantly, the synergistic effect was remarkably augmented by 16.3%, 4.1%, and 14.3% upon irradiation in RPL36A-knockdown SAS, OEC-M1, and KOSC3 cells, respectively (Figure 5A). Next, we determined the potential effect on radiosensitivity by clonogenic survival assays. SAS and OEC-M1 cells with knockdown or overexpression of RPL36A were exposed to different doses of irradiation (0, 4, or 6 Gy), and then assessed in clonogenic cell survival assays; the transfected KOSC3 cells could not tolerate 7-day incubation after 4 Gy irradiation treatment for the clonogenic survival assay. Figure 5C shows the efficacy of RPL36A overexpression by Western blotting. The number of colonies formed, representing cell survival, was decreased by irradiation in a dose-dependent manner (0, 4, and 6 Gy). Upon irradiation, the colony number was significantly decreased for RPL36A-knockdown SAS and OEC-M1 cells as compared with the control cells (Figure 5B). Consistently, the colony number was increased for RPL36A-expressing cells as compared with the control cells after irradiation exposure (Figure 5D). Taken together, these results confirm that RPL36A expression contributes to radioresistant OSCC cells.

## 4. Discussion

In this study, we compared the prognostic values of all genes expressed in TCGA-OSCC patients with and without radiotherapy. We further identified 36 potential radioresistance-associated gene candidates that were upregulated in cancerous tissues relative to normal tissues. According to the literature search, seven of these potential gene candidates had been previously reported to be involved in radioresistance (Table 1). UBE2T downregulation has been shown to enhance the radiosensitivity of osteosarcoma in vitro and in vivo [24]. FANCG has been identified as being significantly differentially regulated post irradiation in prostate cancer cells [30]. Colorectal cancer cells deficient in the DNA damage repair protein EME1 have shown significantly increased radiosensitivity to ruthenium-arene complex treatment [32]. NUDT1 protein levels in mouse organs could serve as a dose-dependent marker of exposure to ionizing radiation, which is known to induce oxidative stress [35]. GINS2 has been found to be differentially expressed in the radioresistant subpopulation within GBM [37]. A microarray analysis has shown that PABPC1L was upregulated in preoperative radiotherapy patients with rectal cancer [39]. These results support the practical feasibility of the strategy we proposed here for identifying candidate radioresistance-associated genes in cancers. Furthermore, we adopted a strategy to filter out potential radioresistance-associated genes based on transcriptome and survival information. A similar strategy could also be applied to identify other radioresistance-associated molecules or modifications, such as miRNA, lncRNA, or DNA methylation. Given that previous studies as well as our validation results showed that these radioresistance-associated genes are mostly involved in translation or DNA repair pathways, molecules or modifications that are directly or indirectly involved in these pathways are more likely to be radioresistance-associated. Functional analysis could serve as another criterion for identifying radioresistance-associated genes. Furthermore, increasing amounts of multiomic data have become publicly available. These multiomic data could help us to further improve our current strategy. For example, nonsynonymous mutations in radioresistance-associated genes could result in misfunction; hence, patients carrying these mutations may behave more similarly to patients with low expression of radioresistance-associated genes. These kinds of associations can only be identified if both genomic and transcriptomic data are taken into consideration at the same time. Hence, in the future, it is worth investigating how to integrate multiomic data to identify radioresistance-associated genes.

The core molecular pathways and gene ontology analysis revealed that radioresistance-associated genes were primarily involved in translation-related pathways (Figure 1B). Importantly, similar results were observed among the genes positively associated with RPL36A (Figure 3B). Ribosomes are cytoplasmic granules composed of RNA and protein in which protein synthesis occurs. RPL36A, which shares sequence similarity with yeast ribosomal protein L44, belongs to the L44E (L36AE) family of ribosomal proteins (RPs). RPL36A, with a molecular weight of approximately 12 kDa, is mainly involved in the formation of the 60S subunit. Kim et al. found that the overexpression of RPL36A was closely related to cellular proliferation in hepatocellular carcinoma [47]. Li et al. identified a novel 5-gene signature (HOXC10, LOC101928747, CYB561D2, RPL36A, and RPS4XP2) as an independent predictor of prognosis in glioma patients who received radiotherapy [48]. However, the prognostic significance and radioresistance-related function of RPL36A in OSCC cells has not been elucidated. Previous studies have shown that RPs play a critical role in gene translation [49]. Accumulating evidence has revealed that RPs are also dysregulated and are involved in radioresistance in many cancer types. Ribosomal S6 protein kinase 4 (RSK4) plays a pivotal role in promoting cancer stem-like cell properties and radioresistance in esophageal squamous cell carcinoma [50]. The ribosomal protein S6 kinase alpha-3 (RSK2)-mediated degradation of FOXN2 promotes tumorigenesis and radioresistance in lung cancer cells [51]. Ribosomal protein S3 (RpS3) ubiquitination is required for radioresistance-induced signaling in GBM [52]. Collectively, RP-associated proteins might be promising prognostic indicators and new potential therapeutic targets, which, in combination with radiotherapy, could enhance the radiotherapeutic efficacy of cancer treatment.

Irradiation induces G2/M cell cycle arrest, and apoptosis appears to be a universal phenomenon in tumor cells [53]. Moreover, tumor cells are more sensitive to radiation-induced damage when the cell cycle is arrested in the G2/M phase, and G2/M arrest is an indicator of cellular radiosensitivity [54,55]. Our results showed that the number of cells arrested in the G2/M phase (Figure 4) and undergoing apoptosis (Figure 5) were positively correlated with the degree of sensitization to radiation. This finding might represent a plausible explanation for the more radiosensitive phenotype of RPL36A-siRNA-transfected cells. Previous studies have indicated: that radiation-induced DNA damage, G2/M arrest, and apoptosis can be regulated by several pathways, for example, the inactivation of Cdc25 and activation of Wee1 can result in G2/M phase arrest via the disassembly of Cdc2-cyclin B heterodimers [54]; p53 inhibits G2/M progression through the direct repression of genes such as CDK1, cyclin B1, and cdc25 [56]; the MAPK pathway controls G2 progression and regulates the G2 DNA damage checkpoint via RAS activation [57]; and the knockdown of uPAR induces sustained G2/M cell cycle arrest followed by cell death [58]. Accordingly, it will be worthwhile to evaluate whether RPL36A is also involved in these cellular mechanisms or regulates different protein molecules to modulate OSCC radiosensitivity. Nevertheless, the current study uncovered a novel role for RPL36A in radioresistant OSCC, which is mediated by cell cycle and apoptosis pathway. These integrated studies are expected to have tremendous potential for the development of new therapeutic interventions for radioresistant OSCC in the future.

## 5. Conclusions

The current study illustrates a systematic analysis strategy for identifying specific radioresistance-associated genes in a large-scale genomic sequencing dataset and identifies a radioresistance-associated candidate, RPL36A. Two independent cohorts were used to confirm that a higher RPL36A transcript level was significantly associated with a poor prognosis in OSCC patients with radiotherapy. Furthermore, the knockdown of RPL36A increased radiosensitivity by promoting G2/M cell cycle arrest and Chk2 activation followed by augmenting cell cycle dependence of irradiation-induced apoptosis pathway in OSCC cells (Figure 5E). Thus, the current study elucidates the prognostic significance and molecular pathway of a novel radioresistance marker, RPL36A, in OSCC. Accordingly, the identification of radioresistance-associated molecules contributing to a poor prognosis can be extended to clinical trial and facilitate patient consulting to determine personalized medicine treatment to improve the therapeutic outcome.

## Figures and Tables

**Figure 1 cancers-13-05623-f001:**
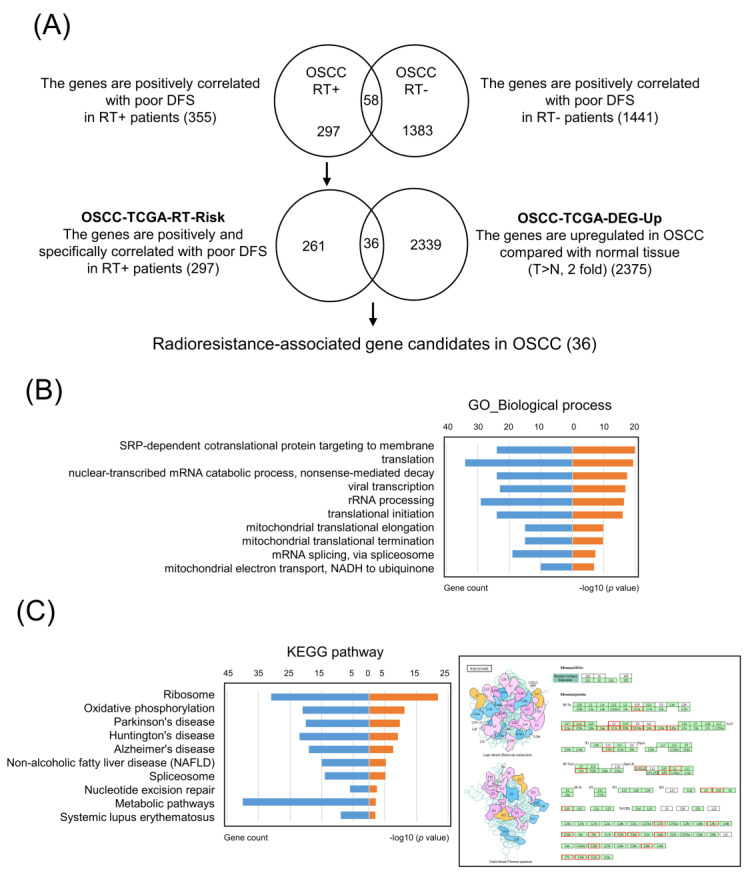
Identification of specific radioresistance-associated and upregulated gene candidates for OSCC from the TCGA dataset: (**A**) Schematic diagrams show the workflow design for profiling radioresistance-associated gene candidates in OSCC via comparative analysis of the TCGA dataset. Enrichment analysis of the 297 genes associated with radioresistance in OSCC; (**B**) significantly enriched GO biological process annotations; (**C**) significantly enriched KEGG pathways (left). KEGG pathway annotations of the ribosome pathway (right). The nodes indicated in red represent genes with high expression associated with poor DFS.

**Figure 2 cancers-13-05623-f002:**
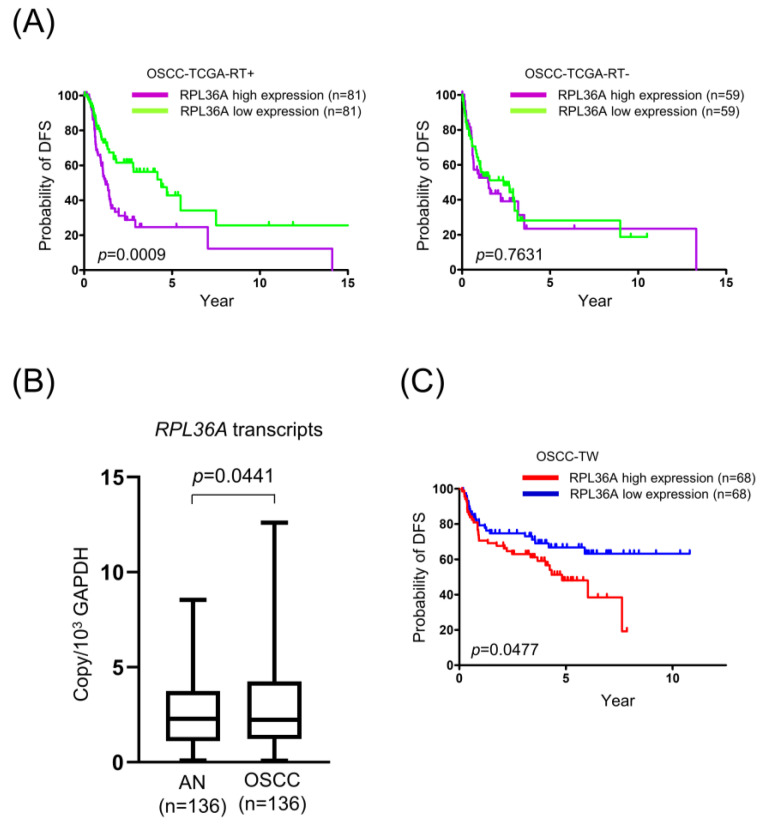
RPL36A overexpression predicts a poor response in the OSCC patients with radiotherapy in the TCGA and Taiwanese cohorts: (**A**) Kaplan-Meier analysis of OSCC patients with high or low RPL36A expression based on a median cutoff point. Plots of the DFS of OSCC patients with radiotherapy (left) and without radiotherapy (right) are shown. The *p*-values were calculated by using log-rank tests; (**B**) RPL36A transcript levels in 136 paired Taiwanese OSCC tissues were determined by qPCR; (**C**) Kaplan-Meier plot showing the 5-year DFS rates for patient subgroups stratified by high versus low RPL36A expression of 51.5% and 67.7, respectively (*p* = 0.0477, log-rank test), in the OSCC-Taiwan dataset.

**Figure 3 cancers-13-05623-f003:**
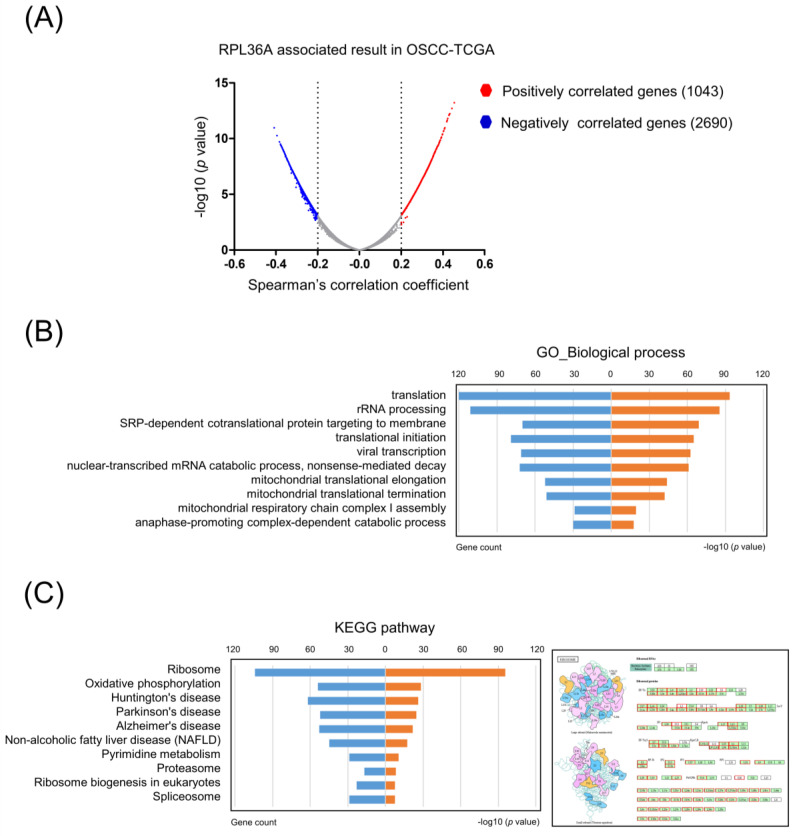
Core molecular pathways and gene ontology annotations of RPL36A-associated genes in OSCC-TCGA: (**A**) Volcano plot of the correlation between the expression of RPL36A and all the other genes in OSCC-TCGA. The x-axis indicates Spearman’s correlation coefficient. The y-axis indicates the adjusted *p*-values plotted in −log10. Red dots indicate the genes positively correlated with RPL36A (Spearman’s correlation coefficient > 0.2). Blue dots indicate the genes inversely correlated with RPL36A (Spearman’s correlation coefficient < −0.2). Enrichment analysis of the 1043 genes positively associated with RPL36A in the OSCC-TCGA dataset; (**B**) significantly enriched GO biological process annotations; (**C**) significantly enriched KEGG pathways (left). KEGG pathway annotations of the ribosome pathway (right). The nodes indicated in red represent genes with expression levels positively correlated with RPL36A.

**Figure 4 cancers-13-05623-f004:**
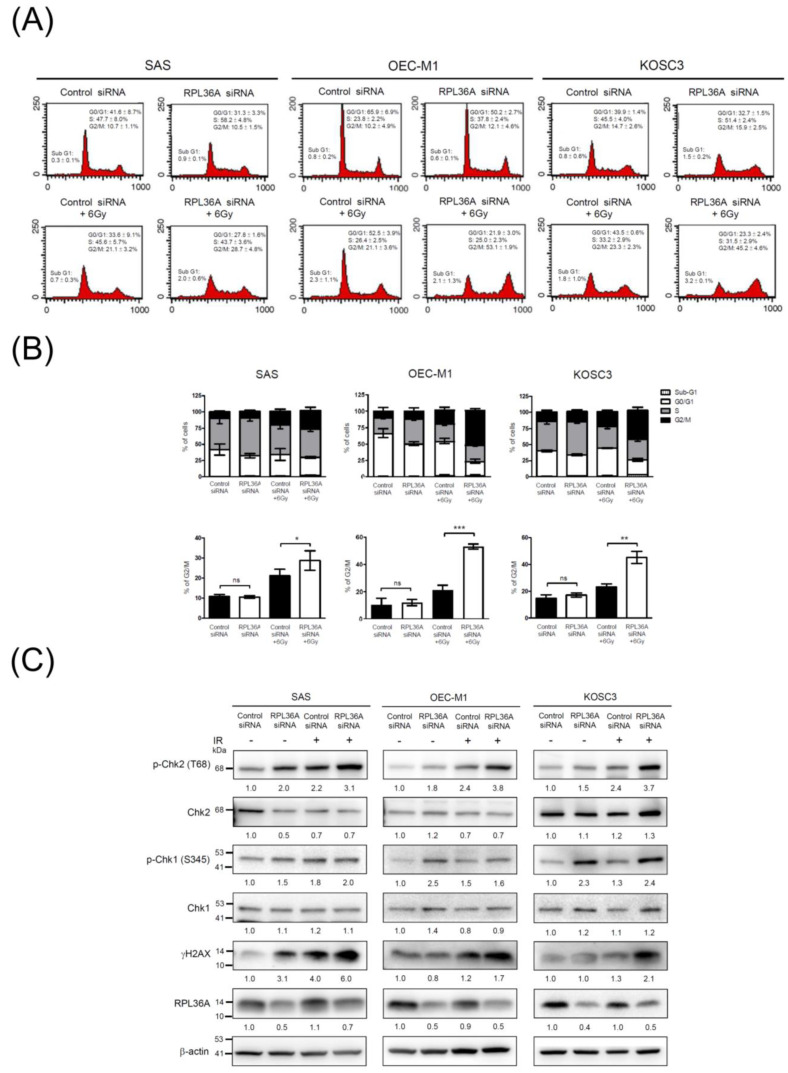
RPL36A depletion sensitizes cells to DNA damage and promotes G2/M cell cycle arrest in response to irradiation: (**A**) SAS, OEC-M1, and KOSC3 cells were transfected with control siRNA and RPL36A-specific siRNA. Cells were simultaneously subjected to 6 Gy irradiation, flow cytometry analysis, and Western blotting. After irradiation for 48 h, the DNA content was determined by PI staining followed by flow cytometry analysis; (**B**) the bar charts show the distribution of the transfected cells in the Sub-G1, G1, S, and G2/M cell cycle phases. Quantification analysis of the cell cycle with G2/M distributions acquired from three independent experiments. Mean values of three independent experiments ± SD are shown. The *p*-values were calculated by using unpaired Student’s *t*-test. ns, nonsignificant, * *p* < 0.05, ** *p* < 0.01, and *** *p* < 0.001; (**C**) total cellular proteins were subjected to Western blotting and analyzed using the indicated antibodies. β-Actin was used as an internal control. The values represent the quantified signals of Western blotting obtained from indicated antibodies and normalized to those of β-actin.

**Figure 5 cancers-13-05623-f005:**
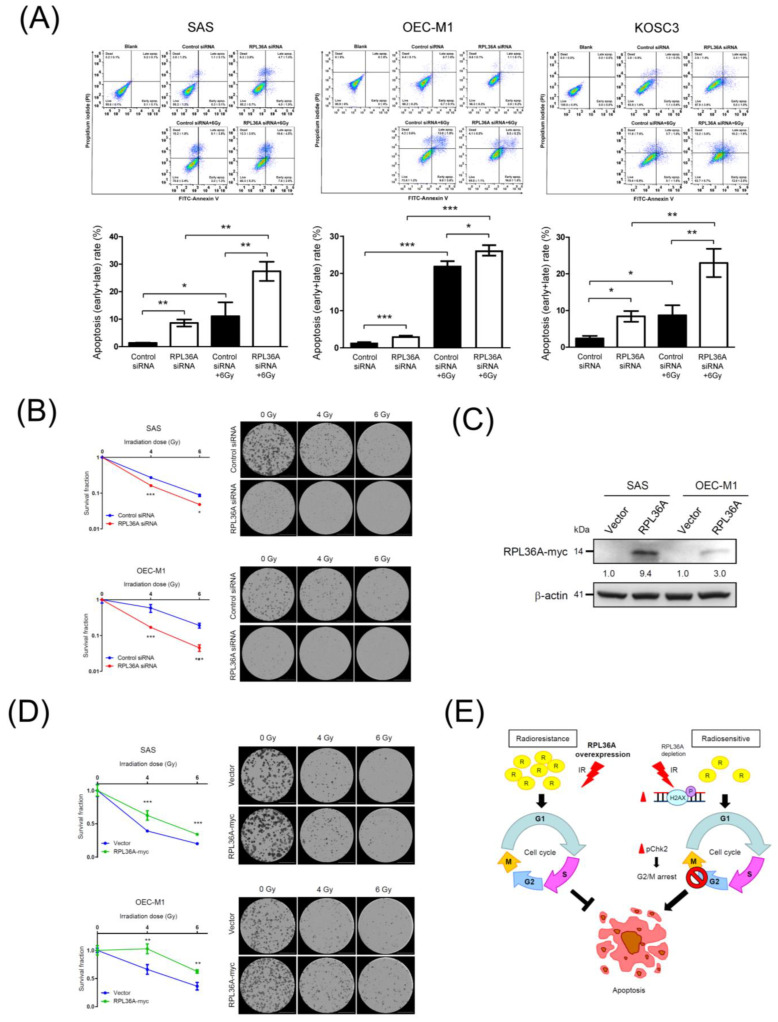
RPL36A depletion augments apoptosis in response to irradiation: (**A**) SAS, OEC-M1, and KOSC3 cells were transfected with control siRNA and RPL36A-specific siRNA. After 6 Gy irradiation of SAS cells for 72 h, OEC-M1 cells for 96 h and KOSC3 cells for 48 h, apoptosis was determined by annexin V-FITC/PI staining followed by flow cytometry analysis. The following four populations were identified: live cells (annexin V-FITC and PI negative), early apoptotic cells (annexin V-FITC positive and PI negative), late apoptotic cells (annexin V-FITC positive and PI positive), and dead cells (annexin V-FITC negative and PI positive) (top). Quantification analysis of apoptosis (early + late) recorded in three independent experiments. Mean values of three independent experiments ± SD are shown. The *p*-values were calculated by using unpaired Student’s *t*-test. * *p* < 0.05 and ** *p* < 0.01 (bottom). Clonogenic survival assays performed with RPL36A-knockdown (**B**) and RPL36A-expressing (**D**) cells after exposure to different doses of irradiation (0, 4, and 6 Gy). (**C**) Western blotting analysis was used to determine RPL36A expression levels in RPL36A-expressing SAS and OEC-M1 cells. β-Actin was used as an internal control. The number of colonies (>50 cells/colony) was counted and analyzed by a Lionheart FX automated microscope. Data are presented as the mean values obtained from three independent experiments. Error bars indicate SD. * *p* < 0.05, ** *p* < 0.01, and *** *p* < 0.001. (**E**) Hypothetical schematic of a regulatory role of RPL36A in developing radioresistance in OSCC. In OSCC cells, elevated RPL36A levels triggered a radioresistance phenotype. Furthermore, under a condition of RPL36A repression, the decreased RPL36A level sensitized cells to DNA damage and induced cell cycle G2/M arrest followed by augmenting irradiation-induced apoptosis pathway, as a consequence of increased OSCC radiosensitivity.

**Table 1 cancers-13-05623-t001:** List of 36 specific radioresistance-associated gene candidates in OSCC.

Gene Name	DFS (*p*-Value)	Dysregulated in HNSCC (Ref.)	Function Related with Radioresistance (Ref.)
HIST1H2BH	1.81 × 10^−4^	No	No
HIST1H4I	6.64 × 10^−4^	No	No
RPL36A	9.47 × 10^−4^	No	Yes [18]
HIST1H3A	1.34 × 10^−3^	No	No
HIST1H3F	1.41 × 10^−3^	Yes [19]	No
ARTN	1.94 × 10^−3^	Yes [20,21]	No
PSMC3IP	4.82 × 10^−3^	No	No
CCDC78	9.17 × 10^−3^	No	No
FBXL6	9.17 × 10^−3^	No	No
CENPH	9.29 × 10^−3^	Yes [22]	No
ACOT7	1.07 × 10^−2^	No	No
UBE2T	1.10 × 10^−2^	Yes [23]	Yes [24]
FAM72B	1.26 × 10^−2^	No	No
TRIML2	1.34 × 10^−2^	Yes [25]	No
HIST1H2BI	1.59 × 10^−2^	No	No
OIP5	1.63 × 10^−2^	No	No
ZWINT	1.75 × 10^−2^	No	No
NAGS	2.01 × 10^−2^	Yes [26]	No
EPPK1	2.30 × 10^−2^	Yes [27]	No
HAUS8	2.43 × 10^−2^	No	No
TMSB10	2.54 × 10^−2^	No	No
FANCG	2.69 × 10^−2^	Yes [28,29]	Yes [30]
HIST1H3I	2.71 × 10^−2^	No	No
EME1	3.04 × 10^−2^	Yes [31]	Yes [32]
GSDMB	3.32 × 10^−2^	No	No
RANBP1	3.51 × 10^−2^	No	No
NFKBIL2	3.66 × 10^−2^	Yes [33]	No
POC1A	3.76 × 10^−2^	No	No
NUDT1	4.15 × 10^−2^	Yes [34]	Yes [35]
GINS2	4.20 × 10^−2^	Yes [36]	Yes [37]
CDC45	4.37 × 10^−2^	Yes [38]	No
PABPC1L	4.56 × 10^−2^	No	Yes [39]
GOLGA8B	4.76 × 10^−2^	No	No
ATHL1	479 × 10^−2^	No	No
SPC25	4.83 × 10^−2^	No	No
CKS2	4.85 × 10^−2^	Yes [26]	No

Abbreviations: OSCC, oral cavity squamous cell carcinoma; DFS, disease-free survival; HNSCC, head and neck squamous cell carcinoma.

**Table 2 cancers-13-05623-t002:** Results of univariate and multivariate analyses of disease-specific survival of 147 OSCC patients with radiotherapy from the TCGA dataset.

	Univariate			Multivariate		
Variables	Hazard Ratio	95% CI	*p-*Value	Hazard Ratio	95% CI	*p-*Value
Age, years ^a^	1.005	0.985–1.025	0.6130	1.014	0.990–1.038	0.2647
Sex						
Male	Reference			Reference		
Female	1.014	0.611–1.684	0.9570	0.811	0.466–1.410	0.4579
Smoking						
No	Reference			Reference		
Yes	0.969	0.591–1.589	0.9010	0.867	0.515–1.458	0.5899
Overall TNM stage						
I–II	Reference			Reference		
III–IV	1.912	0.825–4.429	0.1310	1.500	0.608–3.706	0.3790
Node classification						
*n* ≤ 1	Reference			Reference		
*n* > 1	2.032	1.271–3.246	0.0030 ^b^	2.055	1.237–3.415	0.0054 ^b^
RPL36A						
Low	Reference			Reference		
High	2.144	1.328–3.462	0.0018 ^b^	1.980	1.216–3.224	0.0060 ^b^

Abbreviations: CI, confidence interval; TNM, tumor, node, and metastasis; ^a^, continuous variable; ^b^, statistically significant.

## Data Availability

The data presented in this study are available in Broad GDAC Firehose (https://gdac.broadinstitute.org, TCGA data version 28 January 2016).

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
