# Peer review of "Characterization of Recurrent Relevant Genes Reveals a Novel Role of RPL36A in Radioresistant Oral Squamous Cell Carcinoma"

_cancers, 2021, doi:10.3390/cancers13225623_

Round 1

Reviewer 1 Report

The figures are blurred and difficult to see properly. 

Figure 2B is totally uninformative; can you not plot this in a different manner? A box plot for normal and OSCC would be clearer.

Figure 4 Again the small text is unreadable. The authors conclude that knockdown of RPL36A depletion is activating the G2/M checkpoint. I would argue that the main direct effect seen with RP36A depletion is an increase in DNA damage which is then responsible for the G2 accumulation. The assays employed are not ideal for the purposes required. GH2AX should be analysed by foci quantification; GH2AX is activated by replication stress as well as DNA DSBs, and a WB will not differentiate here. The G2/M checkpoint would be better analysed by phospho Histone H3 analysis.

I think there is a general misunderstanding here regarding radiation sensitivity and the G2/M checkpoint. Activation of the G2/M checkpoint serves to facilitate cell survival by allowing the cell time to repair radiation induced DSBs and is the usual response to DNA damage in normal and malignant cells. If RPL36A knockdown induces G2/M arrest, this does NOT account for increased radiation sensitivity. The data suggests RP36A knockdown increases DNA damage, both at baseline and after IR.

Line 486: I don't understand the logic of why an increase in G2/M arrest leads to an increase in the proportion of apoptotic cells?

The focus on apoptosis as the primary cause of irradiation induced cell death is troubling in this manuscript; the references to support this are selective and dated from over 20 years ago. Radiation induced p53 independent apoptosis is a feature in only a small subset of cancers. The text accompanying your data needs revised accordingly.

Author Response

Reviewer 1

  1. The figures are blurred and difficult to see properly. 

Our response: The authors really appreciate the reviewer’s comment. Following your suggestion, we have replaced all the figures with high-resolution (600 dpi), which are provided as individual TIFF files as well.

  1. Figure 2B is totally uninformative; can you not plot this in a different manner? A box plot for normal and OSCC would be clearer.

Our response: Thanks for reviewer’s comment. We have changed the Figure 2B for a box plot for normal and OSCC in the revised version of manuscript.

  1. Figure 4 Again the small text is unreadable. The authors conclude that knockdown of RPL36A depletion is activating the G2/M checkpoint. I would argue that the main direct effect seen with RP36A depletion is an increase in DNA damage which is then responsible for the G2 accumulation. The assays employed are not ideal for the purposes required. GH2AX should be analysed by foci quantification; GH2AX is activated by replication stress as well as DNA DSBs, and a WB will not differentiate here. The G2/M checkpoint would be better analysed by phospho Histone H3 analysis.

Our response: Thanks for reviewer’s comment. Following your suggestion, we have modified the description about RP36A depletion is an increase in DNA damage which is then responsible for the G2 accumulation and added the quantified results of Western blotting in Figure 4C to elucidate the levels of g-H2AX and Chk2 phosphorylation (Thr68) were remarkably increased in RPL36A-knockdown SAS, OEC-M1 and KOSC3 cells compared with those in the control cells after irradiation treatment.

The Results section of the original version has been revised as follows:

“3.6. RPL36A depletion sensitizes cells to DNA damage and promotes G2/M cell cycle arrest in response to irradiation” (page 12, lines 466-467)

“These results indicated that knockdown of RPL36A in OSCC cells affected the radiosensitive phenotype via sensitizing cells to DNA damage and activated Chk2, thereby leading to G2/M cell cycle arrest.” (page 13, lines 485-487)

“Because the knockdown of RPL36A sensitizes cells to DNA damage, followed by G2/M arrest in OSCC cells upon irradiation, the proportion of sub-G1 cells, which is an indicator of apoptosis, also increased in RPL36A-knockdown cells upon irradiation (Fig. 4).” (page 14, lines 505-506)

The Figure Legend section of the original version has been revised as follows:

“Fig. 4. RPL36A depletion sensitizes cells to DNA damage and promotes G2/M cell cycle arrest in response to irradiation.” (page 14, lines 491)

“The values represent the quantified signals of Western blotting obtained from indicated antibodies and normalized to those of b-Actin.” (page 14, lines 499-500)

“(E) Hypothetical schematic of a regulatory role of RPL36A in developing radioresistance in OSCC. In OSCC cells, elevated RPL36A levels triggered a radioresistance phenotype. Furthermore, under a condition of RPL36A repression, the decreased RPL36A level sensitizes cells to DNA damage and induced cell cycle G2/M arrest followed by augmenting irradiation-induced apoptosis pathway, as a consequence of increased OSCC radiosensitivity.” (page 16, lines 548-550)

  1. I think there is a general misunderstanding here regarding radiation sensitivity and the G2/M checkpoint. Activation of the G2/M checkpoint serves to facilitate cell survival by allowing the cell time to repair radiation induced DSBs and is the usual response to DNA damage in normal and malignant cells. If RPL36A knockdown induces G2/M arrest, this does NOT account for increased radiation sensitivity. The data suggests RP36A knockdown increases DNA damage, both at baseline and after IR.

Our response: Thanks for reviewer’s comment. Following your suggestion, we have modified the description about RP36A knockdown increases DNA damage as described above.

  1. Line 486: I don't understand the logic of why an increase in G2/M arrest leads to an increase in the proportion of apoptotic cells?

Our response: Thanks for reviewer’s comment. Following your suggestion, we have modified the description about G2/M arrest leads to an increase in the proportion of apoptotic cells.

The Abstract section of the original version has been revised as follows:

“Mechanistically, we found that knockdown of RPL36A increased radiosensitivity via sensitizing cells to DNA damage and promotes G2/M cell cycle arrest followed by augmenting the irradiation-induced apoptosis pathway in OSCC cells.” (page 2, lines 53-54)

The Results section of the original version has been revised as follows:

“3.7. RPL36A depletion contributes to radiosensitivity by augmenting irradiation-induced apoptosis.” (page 14, lines 503-504)

  1. The focus on apoptosis as the primary cause of irradiation induced cell death is troubling in this manuscript; the references to support this are selective and dated from over 20 years ago. Radiation induced p53 independent apoptosis is a feature in only a small subset of cancers. The text accompanying your data needs revised accordingly.

Our response: We have deleted the old references (ref.41, 44 and 45 in original version) and added 4 references recently (ref. 43-46 in revised version) to elucidate that radiation induced apoptosis is a feature in many types of cancer, including OSCC and HNSCC as follows:

Head and neck tumor cells treated with hypofractionated irradiation die via apoptosis and are better taken up by M1-like macrophages. Strahlenther Onkol. 2021 Oct 19. doi: 10.1007/s00066-021-01856-4. Online ahead of print.

Radiotherapy induces cell cycle arrest and cell apoptosis in nasopharyngeal carcinoma via the ATM and Smad pathways. Cancer Biol Ther. 2017 Sep 2;18(9):681-693. doi: 10.1080/15384047.2017.1360442.

Cordycepin Enhances Radiosensitivity in Oral Squamous Carcinoma Cells by Inducing Autophagy and Apoptosis Through Cell Cycle Arrest. Cancer Biol Ther. 2017 Sep 2;18(9):681-693. doi: 10.1080/15384047.2017.1360442.

Paraoxonase‑2 (PON2) protects oral squamous cell cancer cells against irradiation‑induced apoptosis. J Cancer Res Clin Oncol. 2015 Oct;141(10):1757-66. doi: 10.1007/s00432-015-1941-2. Epub 2015 Feb 24.

The Results section of the original version has been revised as follows:

“Notably, previous studies have indicated that irradiation can induce apoptosis in many types of cancer, such as osteosarcoma, lung cancer, OSCC and HNSCC [41-46].” (page 14, lines 508-510)

Reviewer 2 Report

Dear authors,

the paper you have presented is suitable for publication in my opinion. I would slightly improve methods, as patients selection is not very clear and consequential (e.g. "Patient populations and clinical specimens" should be first). Minor English check is necessary. 

Kind regards

Author Response

Reviewer 2

Dear authors,

the paper you have presented is suitable for publication in my opinion. I would slightly improve methods, as patients selection is not very clear and consequential (e.g. "Patient populations and clinical specimens" should be first). Minor English check is necessary. 

Kind regards

Our response: Thanks for reviewer’s comment. We have modified the part of 2.4 Patient populations and clinical specimens as follows, and shift this part to 2.1 in revised version of Materials and Methods

“The testing cohort included 136 patients, whose untreated OSCC tumors were primarily managed by surgical resection with subsequent radiotherapy or CCRT were enrolled in the study.” (page 4, lines 116-118)

Our response: We have re-edited the manuscript by American Journal Experts for English check in the revised version of manuscript.

Reviewer 3 Report

An interesting study comparing the prognostic value of  gene expression in patients affected by oral squamous cell carcinoma in patients undergoing or  not undergoing radiotherapy, identifying 297 potential markers of radioresistance and  specifically uncovering a novel role for RPL36A in the radioresistance of OSCC,. The paper has already been reviewed. The article will be eligible to be published after minor revisions:

I would expand the conclusion paragraph better describing the future perspective following this study's results.

Page 3 line 69 you should add: "Although various drugs have been proposed in the systemical or topical management of OSCC, no current therapies seem to have big effects on patient's prognosis" and cite an article such as: "doi: 10.3390/curroncol28040213. and doi: 10.3390/medicina57060563."

Author Response

Reviewer 3

An interesting study comparing the prognostic value of gene expression in patients affected by oral squamous cell carcinoma in patients undergoing or not undergoing radiotherapy, identifying 297 potential markers of radioresistance and specifically uncovering a novel role for RPL36A in the radioresistance of OSCC. The paper has already been reviewed. The article will be eligible to be published after minor revisions:

  1.  I would expand the conclusion paragraph better describing the future perspective following this study's results.

Our response: Thanks for reviewer’s comment. We have added the future perspective in the conclusion paragraph in the revised version as follows:

“Accordingly, the identification of radioresistance-associated molecules contributing to a poor prognosis can be extended to clinical trial and facilitate patient consulting to determine personalized medicine treatment to improve the therapeutic outcome.” (page 17-18, lines 638-641)

  1. Page 3 line 69 you should add: "Although various drugs have been proposed in the systemical or topical management of OSCC, no current therapies seem to have big effects on patient's prognosis" and cite an article such as: "doi: 10.3390/curroncol28040213. and doi: 10.3390/medicina57060563."

Our response: Thanks for reviewer’s comment. We have added the description and these 2 new references (ref.4 and 5) in the revised version of manuscript. (page 3, lines 70-72)

Round 2

Reviewer 1 Report

Thanks for addressing the concerns I raised, I am happy for the paper to be accepted by Cancers.

This manuscript is a resubmission of an earlier submission. The following is a list of the peer review reports and author responses from that submission.

Round 1

Reviewer 1 Report

Major point:

  • Radioresistance is a huge problem for OSCC but is not the main mode of failure for these patients. Looking at the long-term results of RTOG 9501 (Cooper JS et al 2012), twice as many patients fail distantly than they do locoregionally. This has to be acknowledged. 
  • I am unsure if comparing patients who did or did not receive radiation therapy is an adequate method for determination if RPL36A is a major player in radioresistance. There are many confounders to that. The authors MUST perform a multi-variable analysis of other factors that may confound this result. Indications for adjuvant radiation therapy include T3/T4, presence of LVI, presence of PNI, and multiple lymph nodes. These are already bad actors for locoregional recurrence. Moreover, the presence of positive surgical margins and extracapsular extension necessitates the addition of concurrent cisplatin. Therefore, if someone does not receive radiation, it is VERY LIKELY that they do not have high risk disease. We know from the Kian Ang postoperative study that these low risk patients do quite well without the need for radiation therapy. How do we know if these patients who do not get radiation just have low risk disease? This has to be analyzed in the TCGA dataset as well as the Chang Gung dataset. 
  • Cellular damage from ionizing radiation usually mediates mitotic catastrophe, not apoptosis (except with lymphomatous cells). Is this different for OSCC?
  • To ascertain the impact of RPL36A and ionizing radiation, the gold standard of the clonogenic assay MUST be performed. 

Minor point:

  • The introductory line indicates that oral cavity cancer are the most common HNSCC. However, with declines in smoking and rise in HPV related oropharyngeal cancers, I do not believe this to be true anymore. 
  • For the cell cycle analysis, are there any protein markers that the authors think may be mediating the G2/M arrest? Of course, this is an important checkpoint for DNA damage.

Reviewer 2 Report

Thanks for asking me to review this well written and interesting paper, which has clear clinical relevance to the treatment of head and neck squamous cancer. Finding biomarkers of response to treatment is of significant clinical interest for patient selection and treatment escalation.

Nevertheless I have significant concerns regarding the manuscript as it stands which would preclude publication until these were addressed.

RPL36A has been identified via an omics approach utilising TCGA data. Regarding this approach is a p value of 0.05 appropriate to take into account multiple comparisons? The KM curves in fig 2A show a difference in DFS. Has OS been taken into account? Also a multivariate analysis would assess whether the the relationship with RPL36A is independent or merely correlative with other clinicopathological features.

Figure 2B is uninformative and should maybe be a box plot? The joining lines make any assessment of the data impossible.

I am unconvinced by the arguments of figure 3 and find this unhelpful. I'm not convinced that the similarities of the pathways identified between RPL36A and general radioresistance correlated genes are particularly helpful and don't think that this data merits a standalone figure.

My main concern is that RPL36A is a ribosomal protein which has no published links to radiation resistance and the biological plausibility of the association between this protein and radiation resistance is not convincing. Whilst I can see that there may be an important unknown role for this protein in radiation resistance, the supporting mechanistic data is too weak.

The authors need to use more than one siRNA for knockdown. A clonogenic assay in at least 3 cell lines (preferably at least one of them a primary rather than commercial culture) needs to be performed. A gamma H2AX assay would be informative also in response to radiation. The method of assessing G2/M progression is unreliable; I would suggest a mitotic marker such as pHisH3. Apoptosis is not really the way in which cells die from radiation; an assessment of  mitotic catastrophe (IF for GH2AX with a mitotic marker) would be more appropriate. A mechanistic link between RPL36A and the G2/M cell checkpoint would add a lot to the paper's hypothesis and provide much more substantial evidence for their potentially very worthwhile endeavour.